# Ectopic PU.1 Expression Provides Chimeric Antigen Receptor (CAR) T Cells with Innate Cell Capacities Including IFN-β Release

**DOI:** 10.3390/cancers16152737

**Published:** 2024-08-01

**Authors:** Dennis Christoph Harrer, Matthias Eder, Markus Barden, Hong Pan, Wolfgang Herr, Hinrich Abken

**Affiliations:** 1Department of Hematology and Internal Oncology, University Hospital Regensburg, 93053 Regensburg, Germany; wolfgang.herr@ukr.de; 2Leibniz Institute for Immunotherapy, Division of Genetic Immunotherapy, University Regensburg, 93053 Regensburg, Germany; matthias.eder@stud.uni-regensburg.de (M.E.); markus.barden@ukr.de (M.B.); hong.pan@ukr.de (H.P.); hinrich.abken@ukr.de (H.A.)

**Keywords:** transcription factor, CAR, T cell

## Abstract

**Simple Summary:**

The ectopic expression of the master regulator PU.1 could prove detrimental to CAR T cell functionality despite its upregulation of multiple costimulatory receptors on CAR T cells. These data proffer novel insights into CAR T cell biology and highlight the intricate quest for refining CAR T cell functionality.

**Abstract:**

Chimeric antigen receptor (CAR) T cell therapy has achieved extraordinary success in eliminating B cell malignancies; however, so far, it has shown limited efficacy in the treatment of solid tumors, which is thought to be due to insufficient CAR T cell activation. We hypothesized that the transcription factor PU.1, a master regulator of innate cell functionality, may augment pro-inflammatory CAR T cell activation. T cells were engineered with a CEA-specific CAR together with the constitutive expression of PU.1. CAR-redirected T cell activation was recorded for canonical functionality in vitro under conditions of prolonged repetitive antigen exposure. Ectopic PU.1 expression in CAR T cells upregulated the costimulatory receptors CD40, CD80, CD86, and CD70, which, unexpectedly, did not augment effector functions but hampered the upregulation of 4-1BB, decreased IL-2 production, reduced CAR T cell proliferation, and impaired their cytotoxic capacities. Under “stress” conditions of repetitive engagement of cognate tumor cells, CAR T cells with ectopic PU.1 showed reduced persistence, and finally failed to control the growth of cancer cells. Mechanistically, PU.1 caused CAR T cells to secrete IFN-β, a cytokine known to promote CAR T cell attrition and apoptosis. Collectively, PU.1 can polarize the functional capacities of CAR T cells towards innate cells.

## 1. Introduction

Chimeric antigen receptor (CAR) T cells have demonstrated remarkable therapeutic efficacy in patients suffering from B cell malignancies. However, in the large realm of solid tumors, CAR T cell products with durable anti-cancer activity are not yet available [1]. Poor CAR T cell activity against solid tumors has been attributed to the insufficient activation and proliferation of modified T cells, as well as to their limited persistence [2]. Efforts to improve functional CAR T cell performance has spawned CAR T cell products co-expressing additional costimulatory receptors [3,4,5,6], including CD80/CD86-CD28 [3,5], 41BBL-CD137 [6], CD70-CD27 [7], and CD40-CD40L [8], or the constitutive active CD40/TLR4 [9].

In this context, strategies augmenting the signaling pathways that control inflammatory immune cell activation have gained interest. For upregulating costimulatory receptors, such as CD80 and CD86, PU.1 (Purine Rich Box-1) is a crucial key factor that also controls antigen presentation and the induction of type-I interferon [10]. PU.1 plays an important role in early T cell development, while in healthy adults, PU.1-directed actions are largely restricted to the myeloid and B cell compartment [11]. By initiating the expression of pro-inflammatory cytokines, PU.1 confers an inflammatory phenotype to B cells and macrophages [12,13,14]. Collectively, PU.1 is a key regulator that orchestrates genes involved in inflammation and immune cell activation [15].

Given the wide array of costimulatory molecules controlled by PU.1, we hypothesized that PU.1 overexpression may enhance and extend CAR T cell activity through upregulating several costimulatory receptors. Although costimulatory molecules were indeed upregulated, against our expectation, this did not augment CAR T cell expansion and activation but impaired canonical CAR T cell functionality and persistence, which is likely mediated through the release of T cell-toxic IFN-β, presumably as consequence of myeloid differentiation. The results strongly imply optimizing individual costimulatory pathways, rather than improving the entire co-signaling machinery, to prolong CAR T cell functionality.

## 2. Materials and Methods

### 2.1. Cells and Reagents

Peripheral blood mononuclear cells (PBMCs) were obtained by the Lymphoprep centrifugation (Axis-Shield, Oslo, Norway) of blood donated by healthy donors upon informed consent and approval by the institutional review board. PBMCs were frozen and stored at −80 °C. T cells were maintained in RPMI 1640 medium, 1% (*w*/*v*) GlutaMAX (Gibco, ThermoFisher, Waltham, MA, USA), 100 IU/mL penicillin, 100 µg/mL streptomycin (Pan-Biotech, Aidenbach, Germany), 2 mM HEPES (PAA, Palo Alto, CA, USA), and 10% (*v*/*v*) heat-inactivated fetal calf serum (Pan-Biotech, Aidenbach, Germany). CEA-negative 293T cells (ATCC CRL-3216, American Type Culture Collection ATCC, Manassas, VA, USA) and CEA-positive human pancreatic cancer cells BxPC-3 (ATCC CRL-1420) served as target cells [16]. Tumor cells were cultured in DMEM, 1% (*w*/*v*) GlutaMAX (Gibco, ThermoFisher), 100 IU/mL penicillin, 100 µg/mL streptomycin (Pan-Biotech, Aidenbach, Germany), and 10% (*v*/*v*) heat-inactivated fetal calf serum (Sigma-Aldrich, St. Louis, MO, USA).

### 2.2. CAR T Cell Generation

Cryopreserved PBMCs were defrosted and activated on the same day with the agonistic anti-CD3 monoclonal antibody (mAb) OKT-3, the CD28 mAb 15E8, and IL-2 (1000 IU/mL). Recombinant IL-2 (200 IU/mL) was added on days 2, 3, and 4 after activation. Retroviral transduction was performed as previously reported. Four days after activation (day +4), CAR T cells were isolated via magnetic activated cell sorting as previously described. CAR T cells were used for in vitro assays after a 24 h culture period in IL-2-free medium. In all ensuing in vitro assays, IL-2-free medium was employed, and no additional IL-2 was added. The CEA-specific CAR BW431/26scFv-Fc-CD28-ζ has been previously described [17]. The vectors encoding the CEA-specific CAR linked via a P2A-peptide with either GFP or PU.1 (UniProt Q13318_HUMAN) were synthesized by GenScript Biotech (Piscatawy, NJ, USA).

### 2.3. Western Blot Analysis

CAR T cells were lysed (3 × 10^6^ cells per condition) and lysates were electrophoresed by SDS-PAGE in 4–12% (*w*/*v*) Bis-Tris gels under reducing conditions, blotted and probed with the anti-PU.1 antibody (Polyclonal, R&D Systems, Inc., Minneapolis, MN, USA) at 1 µg/mL, and stained by the peroxidase-labeled anti-sheep IgG antibody (R&D Systems) at 1:1000 dilution. Membranes were stripped and re-probed with peroxidase-labeled anti-β-actin antibody (Santa Cruz Biotechnology, Inc., Dallas, TX, USA) at 1:20,000 dilution and visualized by chemoluminescence (ChemiDoc Imaging System, BioRad, Hercules, CA, USA).

### 2.4. Flow Cytometry

For surface staining, cells were incubated with antibodies at 4 °C for 15 min. For intracellular staining of Ki-67, cells were prepared using the “Transcription Buffer” set (BD Biosciences, Franklin Lakes, NJ, USA) for 30 min at 4 °C or for staining of granzyme B and perforin using the “BD Cytofix/Cytoperm” set (BD Biosciences). The viability dye eFluor 780 (ThermoFisher, Waltham, MA, USA) was employed for live/dead discrimination. Fluorescent-minus-one (FMO) controls were used for gating. The goat F(ab’)2 anti-human IgG-PE antibody to detect the CAR was purchased from SouthernBiotech. The following antibodies were purchased from Miltenyi Biotec (Bergisch Gladbach, Germany): FITC-conjugated anti-CD3 (clone BW 264/56), APC-conjugated anti-CD4 (clone VIT4), PE-conjugated anti-CD25 (clone 4E3), APC-conjugated anti-Granzyme B (clone REA 226), APC-conjugated anti-CD70 (clone REA292), and APC-conjugated anti-CD8 (clone BW135/80). The following antibodies were acquired from Biolegend (San Diego, CA, USA): BV421-conjugated anti-CD8 (clone RPA-T8), BV421-conjugated anti-CD3 (clone OKT3), APC-conjugated anti-Perforin (clone B.D48), APC-conjugated anti-human HLA-A/B/C (clone W6/32), APC-conjugated anti-human HLA-DR/DP/DQ (clone Tü39), APC-conjugated anti-CD40 (clone 5C3), APC-conjugated anti-CD54 (clone HA58), PerCP-Cy5.5-conjugated anti-CD86 (IT2.2), PE-conjugated anti-CD80 (clone 2D10), and PerCP/Cyanine5.5-conjugated anti-CD4 (clone RPA-T4). The following antibodies were supplied by BD Biosciences: BV421-conjugated anti-CD137 (clone 4B4-1), BV421-conjugated anti-Ki67 (clone Ki-67), and PE-conjugated anti-CD11a (clone G43-25B). For Annexin V staining, the PE “Annexin V Apoptosis Detection Kit” (BD Biosciences) was used according to the manufacturer’s instructions. For recording proliferation, CAR T cells were labeled with 10 µM “Cell Proliferation Dye eFluor® 450” (ThermoFisher) before stimulation. Immunofluorescence was detected using a BD FACSLyric (BD Biosciences). Data were analyzed using the FlowJo software version 10.7.1 Express 5 (BD Biosciences).

### 2.5. Cytokine Secretion

Target cells were seeded in 96-well round-bottom plates (1 × 10^5^ cells/well) overnight before the addition of CAR T cells (1 × 10^5^ cells/well). Following 48 h of co-culture, IL-2 and IFN-γ in culture supernatants were determined by ELISA as previously reported [18]. Analysis of IFN-β in the supernatant was performed using the rapid bioluminescent human IFN-β ELISA kit (InvivoGen, San Diego, CA, USA).

### 2.6. Cytotoxicity Assay

CAR T cells (0.125-1 × 10^4^ cells/well) were incubated for 24 h together with target cells (each 1 × 10^4^ cells/well) at the indicated effector-to-target ratios. The 2,3-bis(2-methoxy-4-nitro-5-sulphonyl)-5[(phenyl-amino)carbonyl]-2H-tetrazolium hydroxide XTT-based colorimetric assay using the “Cell Proliferation Kit II” (Roche Diagnostics, Mannheim, Germany) was employed to record specific cytotoxicity via the colorimetric monitoring of the reduction of XTT to formazan by viable tumor cells. T cells are not usually capable of metabolizing XTT. The viability of tumor cells was determined as mean values of six wells containing only tumor cells subtracted from the mean background level of wells containing medium only. T cell-mediated formazan formation was determined from triplicates containing T cells in the same number as in the corresponding experimental wells. Cytotoxicity (percent) was calculated as follows: 1—(optical density [OD] [experimental wells—corresponding number of T cells]/OD [tumor cells—medium background]) × 100.

### 2.7. Repetitive Stimulation Assay

GFP-labeled BxPC-3 cells were seeded in 12-well plates (0.1 × 10^6^ cells per well). After 24 h, CAR T cells (0.1 × 10^6^ CAR T cells per well) were added. Three days later (Round 1, R1), T cells and tumor cells were harvested, washed with PBS, and resuspended in 1 mL medium. Then, an aliquot of 100 μL was used for cell counting (live GFP+ tumor cells and live CD3+ T cells) by flow cytometry using counting beads (“CountBright”, ThermoFisher). The remaining 900 μL was added to a fresh 12-well plate with 0.1 × 10^6^ BxPC-3 cells for the second round (R2) of stimulation lasting for four days.

### 2.8. Statistical Analysis

Statistical analysis was performed utilizing GraphPad Prism, Version 9 (GraphPad Software, San Diego, CA, USA). *P* values were determined by Student’s *t* test or the paired *t* test as specified in the figure legend; ns indicates not significant, * *p* ≤ 0.05, ** *p* ≤ 0.01, *** *p* ≤ 0.001, and **** *p* ≤ 0.0001.

## 3. Results

### 3.1. Ectopic PU.1 Expression Upregulates Costimulatory Receptors in CAR T Cells

To generate CAR T cells with constitutive ectopic PU.1 expression, we engineered a retroviral expression cassette encoding a carcinoembryonic antigen (CEA) targeting CD28- ζ CAR and linked the CAR expression cassette to PU.1 (aCEA-28ζ-PU.1) via a P2A peptide. A vector for the CAR with co-expressed GFP served as control (aCEA-28ζ-GFP) (Figure 1A). Upon transduction, aCEA-28ζ-PU.1 CAR and aCEA-28ζ-GFP CAR were expressed at similar levels on the T cell surface (Figure 1B). CAR T cells were further enriched by magnetic cell sorting (MACS) to a purity > 90% CAR^+^ cells for further analyses (Figure 1B). PU.1 is not physiologically expressed by mature peripheral blood T cells; ectopic PU.1 expression in transduced CAR T cells was monitored by Western blot analysis (Figure 1C). CAR T cells engineered with the aCEA-28ζ-P2A-PU.1 vector showed tangible PU.1 levels; no PU.1 was detected in control aCEA-28ζ-GFP CAR T cells.

The bioactivity of ectopically expressed PU.1 in CD4^+^ and CD8^+^ CAR T cells was addressed by monitoring PU.1 target genes before and after 2 days of stimulation of anti-CEA CAR T cells with CEA^+^ BxPC-3 pancreatic tumor cells. While MHC-I expression was augmented both at baseline and after antigen-specific stimulation in aCEA-28ζ-PU.1 CAR T cells, MHC-II expression was substantially enhanced after CAR stimulation (Figure 1D). In comparison to aCEA-28ζ-GFP CAR T cells, we observed an enhanced upregulation of costimulatory receptors, including CD40, CD70, CD80, and CD86, in aCEA-28ζ-PU.1 CAR T cells following CAR stimulation (Figure 1D). The effect was recorded in both CD8^+^ and CD4^+^ T cells. Finally, the ectopic expression of PU.1 induced a pronounced upregulation of the adhesion molecule CD54 and the myeloid marker CD33 as compared to control CAR T cells, while the expression of the corresponding ligand CD11a did not significantly change (Figure 1D). Collectively, the ectopic expression of PU.1 upregulates MHC class I and II molecules, costimulatory receptors, and the adhesion molecule CD54 in CAR T cells following antigen-specific stimulation.

### 3.2. Ectopic PU.1 Expression Is Accompanied by Decrease in IL-2 Secretion and CAR T Cell Proliferation

We asked whether a PU.1-mediated upregulation of costimulatory receptors translates into an enhanced activation and functionality of CAR T cells. To this end, we monitored CD25 expression in aCEA-28ζ-PU.1 CAR T cells in comparison to control CAR T cells after co-culture with CEA^−^ 293T cells and CEA^+^ BxPC-3 pancreatic cells. The magnitude of CAR-driven CD25 upregulation in CD8^+^ and CD4^+^ CAR T cells was not affected by ectopic PU.1 expression (Figure 2A). Also, aCEA-28ζ-PU.1 and control aCEA-28ζ-GFP CAR T cells released similar amounts of IFN-γ. No spontaneous background cytokine release was observed. However, CAR T cells with PU.1 expression secreted significantly less IL-2 as compared to control CAR T cells without PU.1 upon co-culture with CEA^+^ BxPC-3 cells (Figure 2B). Given the relevance of IL-2 in sustaining T cell proliferation, we interrogated the proliferative capacity of aCEA-28ζ-PU.1 CAR T cells in comparison to aCEA-28ζ-GFP CAR T cells after a five-day stimulation period with BxPC-3 cells. Ectopic PU.1 expression also decreased the proliferation of both CD8^+^ and CD4^+^ CAR T cells compared to CAR T cells with GFP as the control (Figure 2C); the observation is in accordance with inferior IL-2 secretion by PU.1^+^ CAR T cells. In aggregate, the PU.1-driven upregulation of costimulatory receptors did not accompany augmented CAR T cell activation. On the contrary, the ectopic expression of PU.1 impaired IL-2 secretion and decreased the proliferative capacity of CAR T cells.

### 3.3. Ectopic PU.1 Expression Diminishes the Cytotoxic Capacities of CAR T Cells

We asked whether the cytotoxic capacity of CAR T cells was affected by PU.1 expression and monitored the upregulation of CD137, a surrogate maker for cytotoxicity associated with costimulation, in aCEA-28ζ-PU.1 CAR T cells compared to control CAR T cells after co-culture with CEA- 293T and CEA+ BxPC-3 cancer cells. The magnitude of CAR-triggered CD137 upregulation after antigen-specific stimulation was significantly lower in aCEA-28ζ-PU.1 CAR T cells as compared to control aCEA-28ζ-GFP CAR T cells without transgenic PU.1; this was observed in both CD4+ and CD8+ CAR T cells (Figure 3A). The CAR-triggered cytotoxicity of aCEA-28ζ-PU.1 CAR T cells with PU.1 was recorded in comparison to control aCEA-28ζ-GFP CAR T cells. Across different effector-to-target-cell ratios, aCEA-28ζ-PU.1 CAR T cells exhibited an inferior killing capacity towards BxPC-3 cells as compared to aCEA-28ζ-GFP CAR T cells (Figure 3B), while no background cytotoxicity against 293T cells was observed. The reduced CAR-redirected killing was not due to reduced levels of cytotoxic molecules, since no major difference in the levels of granzyme B and perforin between aCEA-28ζ-PU.1 and control aCEA-28ζ-GFP CAR T cells was recorded (Figure 3C). In sum, the ectopic expression of PU.1 diminishes CAR-triggered cytotoxicity while preventing the upregulation of costimulatory molecules, such as CD137, which is required to unfold optimal CAR T cell cytotoxicity.

### 3.4. Ectopic PU.1 Reduces Functional Persistence of CAR T Cells

We addressed the functional persistence of CAR T cells with and without ectopic PU.1 using an in vitro “stress-test” based on repetitive challenge with CAR cognate CEA+ cancer cells. While control aCEA-28ζ-GFP CAR T cells expanded in a CAR-triggered fashion in response to sequential antigen stimulation, aCEA-28ζ-PU.1 CAR T cells did not proliferate but entered a contraction phase resulting in fewer surviving CAR T cells upon CAR stimulation as compared to control CAR T cells (Figure 4A). Control aCEA-28ζ-GFP CAR T cells were able to eliminate all BxPC-3 cells in both rounds of stimulation, whereas the cytotoxic capacity of aCEA-28ζ-PU.1 CAR T cells was markedly compromised (Figure 4A). Mechanistically, the ectopic expression of PU.1 rendered both CD8+ and CD4+ CAR T cells more prone to apoptosis in response to CAR stimulation as assessed by annexin V staining (Figure 4B). Staining for Ki-67 in response to stimulation with BxPC-3 cells revealed that aCEA-28ζ-PU.1 CAR T cells suffered from an inferior proliferative capacity compared to control aCEA-28ζ-GFP CAR T cells (Figure 4C). We concluded that CAR T cells with ectopic PU.1 expression exhibited a reduced functional persistence predicated on CAR T cell attrition in response to sequential antigen stimulation, decreased cytotoxicity, enhanced apoptosis, and an insufficient proliferative capacity, although costimulatory receptors like CD40, CD70, CD80, and CD86 were increased.

Generally, PU.1 facilitates the expression of genes governing the interface of innate and adaptive immunity, such as IFN-β, along with a variety of costimulatory receptors [10,15]. We previously highlighted IFN-β secretion as an avenue to confer self-limiting activity on CAR T cells products by restriction of CAR T cell survival and proliferation [17]. Whereas CAR T cells physiologically did not spontaneously produce IFN-β, aCEA-28ζ-PU.1 CAR T cells released substantial amounts of IFN-β following CAR stimulation through BxPC-3 cell engagement as compared to control CAR T cells without PU.1 (Figure 4D). We assume that IFN-β release by aCEA-28ζ-PU.1 CAR T cells is a key factor limiting the functional persistence of CAR T cells with ectopic PU.1 expression.

## 4. Discussion

Optimizing appropriate costimulation remains a challenging task to propel modern CAR T cell therapy to the next level [19]. Previous efforts have largely been clustered around introducing additional costimulatory receptors into CAR T cells [3,5,6] or repurposing inhibitory receptors into activating molecules based on the switch receptor design [20,21,22,23]. Here, we followed the approach of boosting CAR T cell costimulation by the ectopic expression of the key myeloid transcription factor PU.1 in CAR T cells. PU.1 is a major trigger in upregulating multiple costimulatory receptors, such as CD80 and CD86, both on γδ T cells and on conventional αβ T cells [24]. PU.1 has also attracted particular interest due to the potential of human γδ T cells to acquire antigen-presenting capacities upon stimulation. Thus, we aimed to augment CAR T cell costimulation by artificial ectopic PU.1 expression. In this context, we demonstrate that the expression of PU.1 in CAR T cells eventuated in the upregulation of multiple costimulatory receptors, such as CD80, CD86, CD70, and CD40, providing proof of concept that ectopic PU.1 expression can alter the phenotype of CAR T cells.

Unexpectedly, ectopic PU.1 expression did not improve CAR T cell functionality but impaired IL-2 secretion, proliferation, cytotoxicity, and functional persistence, despite the augmented expression of a panel of costimulatory receptors. First, we excluded that ectopic PU.1 expression might lead to excessive stimulation and activation; CAR-triggered CD25 upregulation was not increased; however, CD137 levels following the antigen encounter were reduced. On the grounds of the relevance of PU.1 to the functionality of plasmacytoid dendritic cells [10,25], which feature a particular aptitude for secreting large quantities of type-I IFN upon activation [26], we demonstrated that ectopic PU.1 expression evoked CAR-triggered IFN-β release by CAR T cells. While IFN-α and IFN-β have similar functionality, IFN-β shows the highest receptor affinity, resulting in stronger IFN-driven effects [27]. Previously, we demonstrated that type-I IFN augmented apoptosis and confined the anti-tumor activity of CAR T cells during repetitive rounds of antigen stimulation [17], which is similar to CAR T cells with ectopic PU.1 expression. Moreover, IFN-β was reported to cause pronounced attrition of CAR T cells, leading to inferior survival of tumor-bearing mice treated with CAR T cells and IFN-β [28]. In aggregate, PU.1-mediated IFN-β secretion from activated CAR T cells is likely a triggering factor rationalizing the curbed functional persistence and limited overall functionality of aCEA-28ζ-PU.1 CAR T cells. Moreover, PU.1-mediated IFN-β secretion, as well as upregulation of a variety of myeloid-associated molecules, such as CD40 and CD33, makes a partial differentiation into an innate phenotype likely.

## 5. Conclusions

The ectopic expression of the myeloid master regulator PU.1 upregulates multiple costimulatory receptors on CAR T cells; however, it induces IFN-β, compromising effector cell functionality. These data provide novel insights into the flexibility of CA-triggered T cell function and revealed for the first time distinct phenotypic and functional traits in response to ectopic PU.1 expression in T cells.

## Figures and Tables

**Figure 1 cancers-16-02737-f001:**
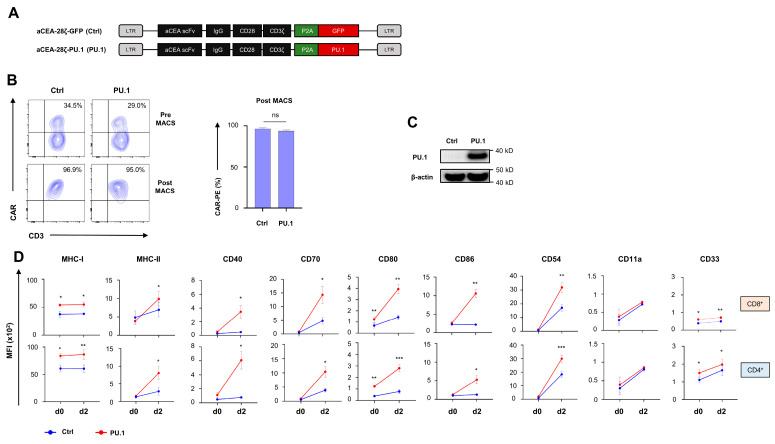
Ectopic expression of PU.1 upregulates costimulatory receptors on CAR T cells. (**A**) Schematic depiction of CAR and PU.1 expression constructs. (**B**) CAR expression by aCEA-28ζ-GFP CAR T cells (Ctrl) and aCEA-28ζ-PU.1 CAR T cells (PU.1) was detected by staining with a phycoerythrin (PE)-labeled goat anti-IgG antibody before (upper panels) and after (lower panels) magnetic cell separation (MACS). The anti-IgG antibody detects the common IgG1 CH2-CH3 spacer domain of the CAR. One representative donor out of six is shown. Data represent means ± SEM of six donors; *p* values were calculated by Student’s *t* test, ns: not significant. (**C**) Western blot showing PU.1 protein expression in aCEA-28ζ-GFP CAR T cells (Ctrl) and aCEA-28ζ-PU.1 CAR T cells (PU.1) five days after retroviral transduction. One representative donor out of six donors is shown. (**D**) Flow cytometric analysis of PU.1 target genes MHC-I and MHC-II, costimulatory receptors, adhesion molecules, and the myeloid marker CD33 in CD8^+^ (upper panels) and CD4^+^ (lower panels) CAR T cells at baseline (d0) and 48 h after stimulation with CEA+ BxPC-3 pancreatic carcinoma cells (d2). Data represent geometric means ± SEM of four donors; *p* values were calculated by paired *t* test. * indicates *p* ≤ 0.05, ** indicates *p* ≤ 0.01, *** indicates *p* ≤ 0.001 and ns indicates not significant.

**Figure 2 cancers-16-02737-f002:**
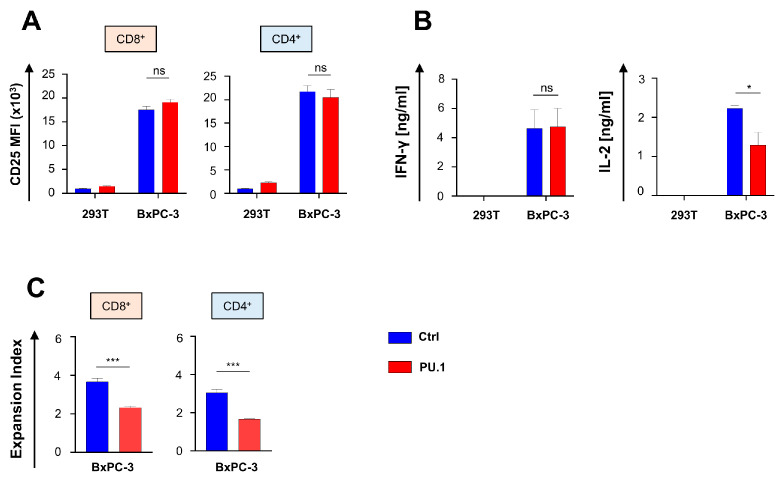
Ectopic expression of PU.1 impairs IL-2 secretion and proliferation of CAR T cells. (**A**) CAR-triggered CD25 upregulation in CD8^+^ and CD4^+^ T cells 48 h after co-incubation with CEA^+^ BxPC-3 cells compared with CEA^−^ 293 T cells. (**B**) CAR-triggered secretion of IFN-γ and IL-2 by CAR T cells incubated with BxPC-3 cells and 293T cells, respectively, after 48 h as determined by ELISA. (**C**) Expansion index of CD8^+^ and CD4^+^ CAR T cells was determined by staining with the “Cell Proliferation Dye eFluor^®^ 450” and co-incubation with BxPC-3 cells for five days. (**A**–**C**) Data represent means ± SEM of four donors; *p* values were calculated by Student’s *t* test; * *p* ≤ 0.05; *** *p* ≤ 0.001; ns, not significant.

**Figure 3 cancers-16-02737-f003:**
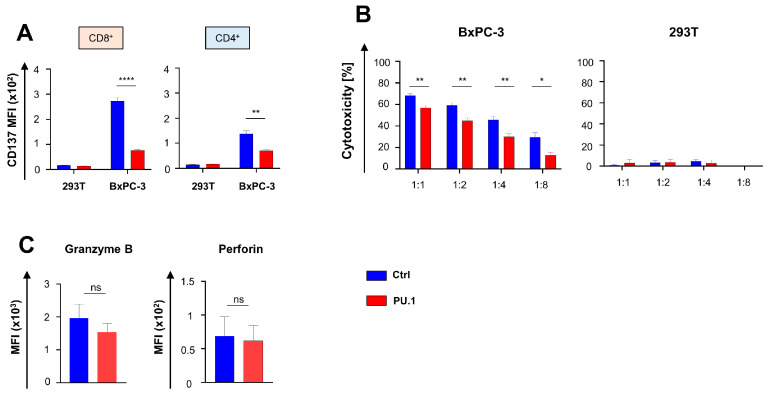
Ectopic PU.1 expression diminishes cytotoxicity of CAR T cells. (**A**) CAR-triggered CD137 upregulation in CD8^+^ and CD4^+^ T cells 48 h after co-incubation with BxPC-3 cells and 293 T cells. (**B**) Cytotoxicity of CAR T cells after a 24-h co-culture with CEA^+^ BxPC-3 cells (left panel) and CEA- 293T cells (right panel) was determined by co-incubation at the indicated effector to target ratios and cytotoxicity recorded by an XTT-based colorimetric assay. (**C**) Flow cytometric analysis of granzyme B and perforin levels in CD8^+^ CAR T cells at baseline after retroviral transduction. (**A**–**C**) Data represent means ± SEM of four donors; *p* values were calculated by Student’s *t* test; * *p* ≤ 0.05; ** *p* ≤ 0.01; **** *p* ≤ 0.0001; ns, not significant.

**Figure 4 cancers-16-02737-f004:**
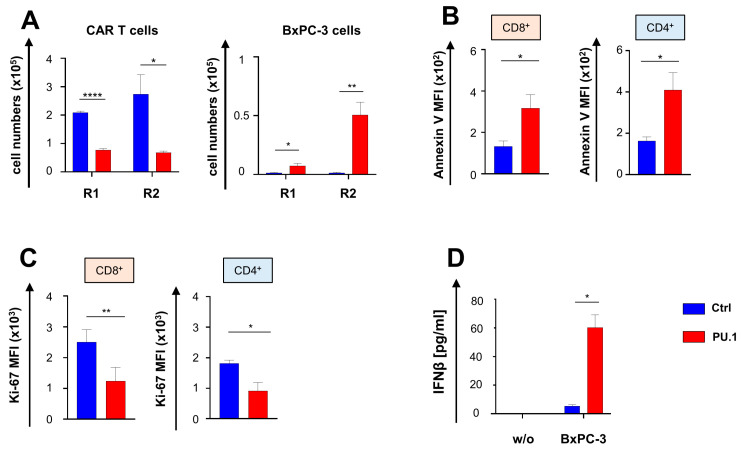
Ectopic expression of PU.1 reduces functional persistence of CAR T cells. (**A**) CAR T cells (1 × 10^5^ CAR T cells) were stimulated twice (R1 and R2) with GFP-labeled CEA^+^ BxPC-3 cells (1 × 10^5^ tumor cells). At the end of each round, live CD3^+^ CAR^+^ T cells (left panel) and BxPC-3 cells (right panel) were counted by flow cytometry. Data represent means ± SEM of five donors; *p* values were calculated by Student’s *t* test; * *p* ≤ 0.05, ** *p* ≤ 0.01; **** *p* ≤ 0.0001; ns, not significant. (**B**) CD8^+^ and CD4^+^ CAR T cells with PU.1 expression were co-incubated with BxPC-3 cells (0.5 × 10^5^ tumor cells) for 72 h and Annexin V staining recorded by flow cytometry. Data represent means ± SEM of four donors; *p* values were calculated by Student’s *t* test; * *p* ≤ 0.05. (**C**) CD8^+^ and CD4^+^ CAR T cells with and without PU.1 were co-incubated with BxPC-3 cells for 72 h and Ki-67 expression determined by flow cytometry. Data represent means ± SEM of three donors; *p* values were calculated by Student’s *t* test; * *p* ≤ 0.05; ** *p* ≤ 0.01. (**D**) IFNβ released into the supernatant by CAR T cells incubated with BxPC-3 cells after 48 h as determined by ELISA. Data represent means ± SEM of three donors; *p* values were calculated by Student’s *t* test; * *p* ≤ 0.05.

## Data Availability

The data of this study are available from the corresponding author upon request.

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
