# Peer review of "Ectopic PU.1 Expression Provides Chimeric Antigen Receptor (CAR) T Cells with Innate Cell Capacities Including IFN-β Release"

_cancers, 2024, doi:10.3390/cancers16152737_

Round 1

Reviewer 1 Report

Comments and Suggestions for Authors

The authors investigated a novel approach to enhance CAR-T therapy by engineering CAR-T cells to express PU.1. However, the negative results presented in this paper diminish the potential clinical translational significance of this approach. Additionally, the writing and presentation of the paper are not well organized. Here are a few suggestions to strengthen this paper:

  1. T cells should be maintained in medium with a low dose of IL-2 (20 U/mL).
  2. Donor-matched activated untransduced T cells should serve as an additional negative control for most of the experiments.
  3. The expression levels of CEA in 293-CEA and BxPC-3 cells should be validated by Western blotting.
  4. In the CAR-T cell killing assay, the effector-to-target ratio should range from 1:10 to 10:1 to avoid overlooking potential differences in killing ability.
  5. Several typos need to be corrected, e.g., "effec-tor" in line 277 and "cy-tometric" in line 278.
  6. There are some mislabelings throughout the paper. For example, in Figure 4, the labels R1 and R2 (round 1 and round 2) should be explained in the figure legend. There is no information about Figure 4.D, and in Figure 4.C, the y-axis is labeled as Ki-67 by mistake.

Author Response

First of all, we would like to thank the reviewer for taking the time to review our manuscript. Please find a detailed point to point response below:

Reviewer 1:

  1. T cells should be maintained in medium with a low dose of IL-2 (20 U/mL).

T cells are supplemented with 1000U/ml IL-2 (two days) and 500U/ml Il-2 (one day during the transduction process). One day before in vitro assays, T cells are depleted from IL-2, to enable IL-2 measuring via ELISA. During the in vitro assays no IL-2 is added which reflects the clinical situation with the commercially available CAR T cell products where no IL2 is added after CAR T cell infusion. Hence, it is the standard in our laboratory to perform in vitro assays without concomitant use of IL-2.

  1. Donor-matched activated untransduced T cells should serve as an additional negative control for most of the experiments.

In this paper, we wanted to investigate the effect on PU.1 expression on CAR T cell performance. Untransduced T cells are generally used to show the influence on allo reactivity. Usually, the allo reactivity is negligible and untransduced T cells do not exert any effect against target cells, so we do not rely on this negative control. For illustration, we included here the killing assay with untransduced T cells showing no activity against bxPC3 target cells.

  1. The expression levels of CEA in 293-CEA and BxPC-3 cells should be validated by Western blotting.

293T cells are not CEA-293T but are not expressing CEA serving as a control cell line. CEA expression in 293T cells and BxPC3 cells was previously documented by FACS from our group.

Harrer, D.C., Schenkel, C., Bezler, V., Kaljanac, M., Hartley, J., Barden, M., Pan, H., Holzinger, A., Herr, W., Abken, H. CAR Triggered Release of Type-1 Interferon Limits CAR T-Cell Activities by an Artificial Negative Autocrine Loop. Cells 2022, 11. https://doi.org/10.3390/cells11233839.

We highlighted the status of CEA expression in the methods section.

  1. In the CAR-T cell killing assay, the effector-to-target ratio should range from 1:10 to 10:1 to avoid overlooking potential differences in killing ability.

The depicted effector to target range shows significant difference in killing capacity at all teste effector to target ratios. At higher effector to target ratios, usually all target cells are killed and at lower effector to target ratios killing capacity is little. We depicted the optimal effector to target range for CAR T cell constructs as judged by our experience.

  1. Several typos need to be corrected, e.g., "effec-tor" in line 277 and "cy-tometric" in line 278.

We apologize for our negligence and corrected all typos.

  1. There are some mislabelings throughout the paper. For example, in Figure 4, the labels R1 and R2 (round 1 and round 2) should be explained in the figure legend. There is no information about Figure 4.D, and in Figure 4.C, the y-axis is labeled as Ki-67 by mistake.

We apologize for our negligence and put the correct figure legend under Figure 4. 

Reviewer 2 Report

Comments and Suggestions for Authors

CAR-T therapy is a very promising method for treating cancer, although currently there are limitations to its use in the treatment of solid tumors. The purpose of this study was to increase the activation of the CAR-T, but the obtained effect was opposite. The insufficient efficiency of the CAR-T may be associated with the depletion of T cells as a result of their hyper activation. In general, the methods used are not described in sufficient detail; there are inconsistencies in their description in various parts of the manuscript.

There are a number of comments on the work:

11.      It is not clear how T cells were maintained, whether the T cells were cultured without IL-2 or any other interleukin. Did the authors notice the decreased proliferation of PU.1 CAR-T in culture?

22.    The XTT-based colorimetric assay is not suitable for measuring CAR T cell cytotoxicity. It is not direct method. Typically, LDH analysis is used for this purpose. Various assay conditions have been mentioned in the figure 3B legend and in the Materials and Methods. It is unclear whether the incubation time was 24 or 48 hours? The viability formula uses the parameter "appropriate number of T cells". How was this measured? How did the authors differentiate between viable tumor cells and CAR-T cells in this assay? In the section 2.6 of Materials and Methods it is mentioned that CAR T cells were used at amount of 10 × 104 cells/well, while the number of target cells were 1 × 104 cells in each well. This E:T ratio is missing from figure 3B.

33.       According to the cytotoxicity assay about 30 percent of tumor cells are viable after incubation with control CAR-T in a 1:1 ratio. In the test of functional persistence of CAR T cells control group eliminated all the target cells. The question arises of the relevance of the techniques used.

44.       The figure 4 legend does not correspond to the content of the figure.

Comments on the Quality of English Language

Spelling check is required.

Author Response

Reviewer 2:

  1. It is not clear how T cells were maintained, whether the T cells were cultured without IL-2 or any other interleukin. Did the authors notice the decreased proliferation of PU.1 CAR-T in culture?

We apologize, and clarified the use of IL2 in the methods section: Recombinant IL-2 (200 IU/ml) was added on days 2, 3, and 4 after activation. Retroviral transduction was performed as previously reported. Four days after activation (day +4), CAR T cells were isolated via magnetic activated cell sorting as previously described. CAR T-cells were used for in vitro assays after a 24 h culture period in IL-2 free medium. In all ensuing in vitro assay, IL-2 free medium was employed, and no additional IL-2 was added. 

Indeed, we detected a decreased proliferation of PU.1 CAR T cells in culture fitting nicely with our proliferation data: The figure depicts the cell count after cell culture right before in start of in vitro assays. As our proliferation data are unequivocal we opted to not include the cell culture data in our manuscript.

  1. The XTT-based colorimetric assay is not suitable for measuring CAR T cell cytotoxicity. It is not direct method. Typically, LDH analysis is used for this purpose. Various assay conditions have been mentioned in the figure 3B legend and in the Materials and Methods. It is unclear whether the incubation time was 24 or 48 hours? The viability formula uses the parameter "appropriate number of T cells". How was this measured? How did the authors differentiate between viable tumor cells and CAR-T cells in this assay? In the section 2.6 of Materials and Methods it is mentioned that CAR T cells were used at amount of 10 × 104 cells/well, while the number of target cells were 1 × 104 cells in each well. This E:T ratio is missing from figure 3B.

We apologize for our negligence, and harmonized the information in the methods section and the Figure legend. The incubation time is 24 hours. The XTT assay has been used in the laboratory of Hinrich Abken, a pioneer of CAR T cell research, for decades:

Hombach, A.A., Rappl, G., Abken, H. Blocking CD30 on T Cells by a Dual Specific CAR for CD30 and Colon Cancer Anti-gens Improves the CAR T Cell Response against CD30(-) Tumors. Molecular therapy: the journal of the American Society of Gene Therapy 2019, 27, 1825–1835. https://doi.org/10.1016/j.ymthe.2019.06.007.

Harrer, D.C., Schenkel, C., Bezler, V., Kaljanac, M., Hartley, J., Barden, M., Pan, H., Holzinger, A., Herr, W., Abken, H. CAR Triggered Release of Type-1 Interferon Limits CAR T-Cell Activities by an Artificial Negative Autocrine Loop. Cells 2022, 11. https://doi.org/10.3390/cells11233839.

Hombach, A., Wieczarkowiecz, A., Marquardt, T., Heuser, C., Usai, L., Pohl, C., Seliger, B., Abken, H. Tumor-specific T cell activation by recombinant immunoreceptors: CD3 zeta signaling and CD28 costimulation are simultaneously required for efficient IL-2 secretion and can be integrated into one combined CD28/CD3 zeta signaling receptor molecule. Journal of im-munology (Baltimore, Md. : 1950) 2001, 167, 6123–6131. https://doi.org/10.4049/jimmunol.167.11.6123.

We changed the misleading term of appropriate T cells and elaborated on XTT assay in the Methods section:

The 2,3-bis(2-methoxy-4-nitro-5-sulphonyl)-5[(phenyl-amino)carbonyl]-2H-tetrazolium hydroxide XTT-based colorimetric assay using the “Cell Proliferation Kit II” (Roche Diag-nostics, Mannheim, Germany) was employed to record specific cytotoxicity via colometrically monitoring of reduction of XTT to formazan by viable tumor cells. T cells are usually not capable of metabolizing XTT. Viability of tumor cells was determined as mean values of six wells containing only tumor cells subtracted by the mean background level of wells containing medium only. T cell-mediated formazan formation was determined from triplicates containing T cells in the same number as in the corresponding experimental wells. The cytotoxicity (percent) was calculated as follows: 1 – (optical density [OD] [ex-perimental wells – corresponding number of T cells] / OD [tumor cells – medium back-ground]) × 100.

10 × 104 cells/well was a mistake and was corrected in the methods section. All effector to target ratios are depicted in the Figure.

  1. According to the cytotoxicity assay about 30 percent of tumor cells are viable after incubation with control CAR-T in a 1:1 ratio. In the test of functional persistence of CAR T cells control group eliminated all the target cells. The question arises of the relevance of the techniques used.

The XTT assay lasts 24 hours, while the FACS assay is run for 72 hours giving T cells more time to kill target cells. The test for functional persistence is intentionally run for a longer time period to factor in T cell persistence, proliferation and killing capacity.

  1. The figure 4 legend does not correspond to the content of the figure.

We apologized for this mistake. The correct figure legend was inserted.

Reviewer 3 Report

Comments and Suggestions for Authors

The authors have reported ectopic expression of PU.1 in CAR T cells and upregulation of CD40, CD80, CD86 as well as CD70 coreptors. However, the resuts did not enhance effector functions of CAR T cells. These results were associted with IL-2 production decreasing and hampered upregulation of 4-1BB. 

Author Response

We are grateful for the overall approval of our manuscript.